# TRIANGULAR DROPOUT:
# VARIABLE NETWORK WIDTH WITHOUT RETRAINING

## ABSTRACT

One of the most fundamental design choices in neural networks is layer width: it affects the capacity of what a network can learn and determines the complexity of the solution. This latter property is often exploited when introducing information bottlenecks, forcing a network to learn compressed representations. However, such an architecture decision is typically immutable once training begins; switching to a more compressed architecture requires retraining. In this paper we present a new layer design, called Triangular Dropout, which does not have this limitation. After training, the layer can be arbitrarily reduced in width to exchange performance for narrowness. We demonstrate the construction and potential use cases of such a mechanism in three areas. Firstly, we describe the formulation of Triangular Dropout in autoencoders, creating models with selectable compression after training. Secondly, we add Triangular Dropout to VGG19 on ImageNet, creating a powerful network which, without retraining, can be significantly reduced in parameters. Lastly, we explore the application of Triangular Dropout to reinforcement learning (RL) policies on selected control problems.

## 1 INTRODUCTION

In this paper we describe Triangular Dropout, a novel dropout (Srivastava et al., 2014) technique that imparts a powerful property to fully-connected layers: the layer can be reduced in width after training to select a desired size of representation. This enables structured pruning in fully-connected layers such that the pruned network is densely connected. This exposes a trade-off between performance and network width that can be varied without retraining the network. Additionally, constructing and applying Triangular Dropout is straightforward. Triangular Dropout amounts to masking minibatches during training with a binary lower triangular matrix with zeros above the diagonal. After training, the relevant weights can be copied into a physically smaller dense network.

Architecturally, Triangular Dropout represents a shift towards models that can be densely downsized to fit the needs of a particular application. For example, a wide model could be trained once on a computer cluster, and narrower versions would be immediately available for use on personal machines, mobile devices, IoT devices, or other embedded platforms. An image processing pipeline could have its frame-rate or feature descriptor length altered as needed, perhaps to process more samples in exchange for lower performance per sample.

Triangular Dropout is tangent to many areas of research, and includes ideas of compression, structured network pruning, and variable computation. Our objective is not to outperform in any specific domain, but rather to demonstrate a technique that we find has useful properties from several areas and is very easy to implement.

We discuss related works to Triangular Dropout in Section 2, and formally define our mechanism in Section 3. We then investigate some applications of Triangular Dropout in unsupervised learning of latent encodings (Section 4), supervised learning in larger models (Section 5), and reinforcement learning for locomotion control problems (Section 6). Sections 7 and 8 further discuss the characteristics of Triangular Dropout and suggest potential future research directions.

## 2 BACKGROUND

The width of a specific network layer is perhaps most critical in the domain of autoencoding (Rumelhart et al., 1986), in which the width of the encoding layer (along with its activation) describes the amount of information retained in an encoded sample. The smaller the latent space (narrower the encoding layer), the more compressed the learned representation, and therefore, the more lossy the compression performed by the network. There are countless variations of autoencoders which seek to improve on this compression ratio (Theis et al., 2017) or introduce additional properties to the latent variables (Vincent et al., 2008; Rifai et al., 2011).

The degree to which the latent variables are independent is the subject of disentanglement research, which introduces further properties into the latent variables. $\beta$-VAE (Higgins et al., 2017) demonstrates that the gaussian distributions in a VAE (Kingma & Welling, 2014) can be made increasingly orthogonal in exchange for reconstruction accuracy. The independence and ordering of learned features can also be enforced manually by methods such as PCAAE (Pham et al., 2020), which learns one latent variable at a time, progressively widening the latent layer to learn additional features via additional training. A comparison to PCAAE is discussed in detail in Section 4.

Autoencoding makes use of a well-known tradeoff between network size and capability to encourage compression, a relationship that holds in other domains (Amodei & Hernandez, 2019; Kaplan et al., 2020; McCandlish et al., 2018). This is perhaps most well known with regards to ImageNet (Deng et al., 2009), in which historically a clear correlation is seen between network size and performance (summarized nicely in Figure 1 of Tan & Le (2020)).

However, such large models can also be very inefficient in their use of parameters, often learning unnecessary weights or redundant representations. Network pruning is a technique to reduce an over-parameterized network down to essential parameters, recovering a sub-network that is similar or even superior in performance (Blalock et al., 2020; Frankle & Carbin, 2018). In unstructured pruning, the network is sparse, whereas in structured pruning, the sub-network is dense. Triangular Dropout falls into the latter category, but examines fully-connected layers instead of the typical pruning of convolutional elements, as in Anwar et al. (2015) (channels and kernels), Li et al. (2016) (filters), and He et al. (2017) (channels). Triangular Dropout additionally does not require finetuning and is not iterative, as in the above works and many others (Molchanov et al., 2016; Luo et al., 2019), although fine-tuning could of course be applied if desired. In Section 5, we explore dense prunings recovered by Triangular Dropout on the classification head of VGG19, an early ImageNet solution that is well-known to be overparameterized as evidenced by subsequent works with fewer parameters (Iandola et al., 2016; Tan & Le, 2020). This overparameterization and history of use makes it popular for pruning studies, especially on ImageNet (Blalock et al., 2020).

There is also a great deal of relevant work from the conditional computation domain, exploring architectures that dynamically find appropriate sub-networks for a given input (Han et al., 2021). In this approach networks are trained to enable selection of sub-networks, rather than simply pruning the network after initial training. Few approaches examine variable feed-forward layer width, for example Bengio et al. (2015) use reinforcement learning to learn which nodes in a fully-connected layer can be masked out on a per-sample basis. Ba & Frey (2013) achieve a similar effect through deep belief networks creating layer masks. Most methods rely on selective routing through entire sub-branches or sub-modules, i.e. Sabour et al. (2017). There is a related area of work in recurrent networks which adapts computation in the time (rollout) dimension during inference, mainly by choosing where and when to update a hidden state (Graves, 2016; Campos et al., 2017; Jernite et al., 2017). In contrast to these areas, Triangular Dropout specifically seeks dense representations that capture the entirety of the input space, and does so in a non-recurrent manner.

Network size selection has additional repercussions in the deep reinforcement learning (DRL) domain. Works like amplified control (Paulhamus et al., 2019) and robust predictable control (Eysenbach et al., 2021) seek compressed policies to perform tasks, using information bottlenecking (Tishby et al., 2001) to enforce this compression. Robust predictable control demonstrates that compression creates policies that are more risk-averse and potentially more transferable. Such compressability is further related to task complexity measurement, an open problem (see the implications of task complexity discussed in Srivastava et al. (2019)). We explore the application of Triangular Dropout to DRL in Section 6.

## 3   TRIANGULAR DROPOUT

Triangular Dropout is simple to implement in practice. For a fully-connected layer of width $n$, processing a minibatch $x$ of $B$ datapoints, the output $y$ has size $(B, n)$. During training, standard dropout (with probability $p$) multiplies the output element-wise by a random binary mask $\boldsymbol{M}$ of the same size as the output:

$$y = \boldsymbol{M} \odot \alpha(\boldsymbol{x}^T \boldsymbol{W} + \boldsymbol{b}) \quad \text{where} \quad \boldsymbol{M} \sim Bernoulli(p)$$

where $\boldsymbol{W}$ and $\boldsymbol{b}$ are the layer weights and biases, respectively, and $\alpha$ is some activation function. Triangular Dropout has the same form, but the mask $\boldsymbol{M}$ is populated as a binary lower triangular matrix instead of a randomly sampled one:

$$y = \boldsymbol{M}_{Tri} \odot \alpha(\boldsymbol{x}^T \boldsymbol{W} + \boldsymbol{b}) \quad \text{where} \quad \boldsymbol{M}_{Tri} = \begin{pmatrix} 1 & 0 & \dots & 0 \\ 1 & 1 & \dots & 0 \\ \vdots & \vdots & \ddots & \vdots \\ 1 & 1 & \dots & 1 \end{pmatrix}$$

This is visualized in Figure 1. In the case where $B > n$, the rows of $\boldsymbol{M}_{Tri}$ are simply repeated to reach the desired size. In the case where $B < n$, a triangular block matrix may be used. For any $B$, an approximate alternative would be to independently sample rows such that a random number of 1's are followed by 0's.

The implications of a triangular mask can be viewed in different ways. Considering the rows of the output $y$, the mask creates a set of datapoints which have been processed with the full range of layer widths from 1 to $n$. If the batch size is equal to $n$, the $ith$ row represents a training sample in which the effective layer width is $i$. Training now amounts to something like a multitask learning problem, in which the "tasks" are processing data with varying network widths. We have an even distribution of data from all these tasks, and the loss function reflects training over this set of architectures (widths):

$$y = \begin{pmatrix} y_{1,1} & 0 & 0 & \dots \\ y_{1,2} & y_{2,2} & 0 & \dots \\ y_{1,3} & y_{2,3} & y_{3,3} & \dots \\ \vdots & \vdots & \vdots & \ddots \end{pmatrix} \begin{matrix} \rightarrow \text{ effective width 1} \\ \rightarrow \text{ effective width 2} \\ \rightarrow \text{ effective width 3} \\ \\ \end{matrix}$$

Taking this multitask view, we do not re-scale the layer outputs as in normal dropout; we want the downstream network to be able to handle any of the "tasks" (any of the potential widths of this layer).

However, the "tasks" in this view are not independent. Examining the columns of the output, the $jth$ column represents the output of the $jth$ layer node, masked out for the first $j$ rows. Because the mask is triangular, all of the samples which are non-zero at column $j$ are necessarily also non-zero at columns 1 through $j - 1$. That is, parameters relevant to the $jth$ node are only updated for samples in which the previous $j - 1$ nodes (smaller widths) are also updated. This is not true for subsequent nodes: nodes $j + 1$ and onwards may or may not be masked when node $j$ is utilized.

This cascading dependence is what masks Triangular Dropout work. The parameters related to node $j$ can be learned with a guarantee that the preceding $j - 1$ outputs in terms of width are also present. However, they cannot rely as heavily on outputs $j + 1$ and beyond, and are essentially restricted from fully co-optimizing with these outputs.

During test time, it is up to the user to decide on the masking strategy. Providing a mask filled with ones makes the layer approximately equivalent to a typical fully-connected layer, or masking out the last $N - j$ nodes causes the layer to behave as though it has width $j$. To permanently ablate the layer down to some desired width, the relevant model parameters (portions of $\boldsymbol{W}$ and $b$ from this layer and the next) simply need to be copied into a smaller model, as shown in Figure 1.

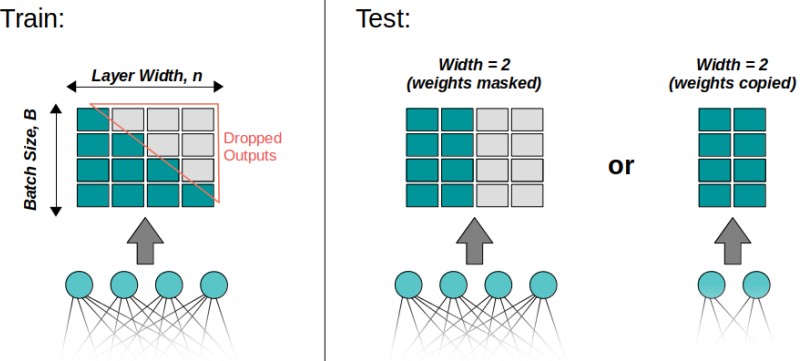

Figure 1: Diagram of Triangular Dropout application in training (Left) and used during test or deployment (Right). During test, a layer width can be simulated by masking the layer output, or realized by copying relevant weights into a narrower architecture.

## 4 VARIABLE COMPRESSION IN AUTOENCODERS

To further detail the properties of Triangular Dropout and compare it to alternative methods, we demonstrate the construction of an autoencoder with a selectable latent size. We train an autoencoder on MNIST (Deng, 2012) with a latent size $n$, and use Triangular Dropout on the latent layer. After training, we can select any level of compression $\leq n$ and receive both a valid encoding and reconstruction. We also train autoencoders on the CelebA dataset (Liu et al., 2015) (image size reduced to 64x64) in order to explore Triangular Dropout on a more difficult autoencoding task.

### 4.1 BASELINES

We compare MNIST results quantitatively (reconstruction loss) and qualitatively (visualization of reconstruction) to two baselines: $n$ autoencoders that are trained for specific latent sizes 1 through $n$, and PCAAE, which trains an autoencoder of size $n$ progressively. PCAAE trains one latent variable at a time and includes a correlation minimization loss term. PCAAE also trains a separate encoder for each latent variable, and a new decoder for each width. This means that, like the set of standard autoencoders, PCAAE requires $n$ training runs.

Compared to standard autoencoders, our expectation is that Triangular Dropout must sacrifice some compression in exchange for the introduction of some one-way dependence among features (the first $i$ features are not very dependent on the $i + 1th$ feature). Conversely, we would not expect a typical autoencoder to have this property. The encoded features from a typical autoencoder, having no encouragement to be independent or disentangled, should only be meaningful when the full feature descriptor is retained.

We would expect that PCAAE further sacrifices reconstruction loss for an even more stringent independence among learned features. However, the features are independent, the reconstructions may still be meaningful when less important features are removed from the final model.

### 4.2 EXPERIMENTS AND RESULTS

A convolutional autoencoder with latent size 32 was trained with PCAAE and Triangular Dropout to reconstruct MNIST digits. The smaller widths of PCAAE were saved off during training. Additionally, standard autoencoders were trained for sizes 1 through 32. We examine the reconstruction loss of the models as the latent width is increased during training and decreased via ablation of the final 32-width model. Triangular Dropout falls into the latter category by design, but has performance similar to the checkpoint models trained at specific sizes. These results are shown in Figure 2. Full architecture and training details are available in the Appendix.

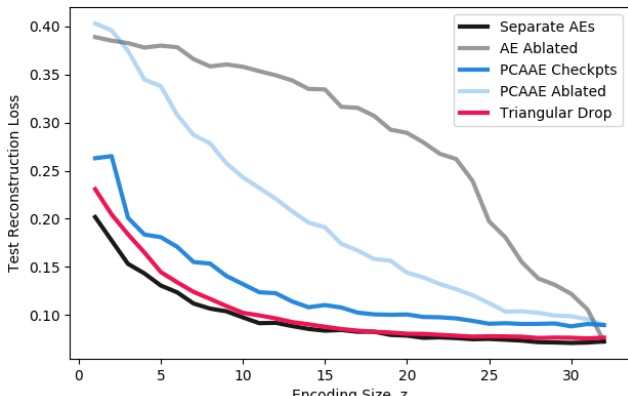

Figure 2: Reconstruction loss versus encoding size for a variety of methods. Triangular Dropout (red) is most similar to independent autoencoders trained at specific encoding sizes (black). PCAAE models produced during progressive training are shown in blue. Light grey and light blue show the reconstruction loss of the full-size standard autoencoder and PCAAE models, respectively, when the latent space is ablated down to a given size.

Notably, our hypotheses regarding feature independence vs reconstruction loss hold true: for a given latent size z, the most performant model (without feature ablation) is a standard autoencoder trained at that width, the least performant is PCAAE which introduces a correlation loss, and somewhere in-between is Triangular Dropout. The baseline models with ablated encodings do not have comparable reconstruction losses unless the ablation is very minor (most latent features retained). Triangular Dropout, trained only at size 32, can be ablated down to very small sizes while only performing slightly worse than individual autoencoders at those sizes. Triangular Dropout roughly encompasses the superset of these autoencoders, while only requiring a single training run.

Figure 3 shows the qualitative reconstruction in each of these cases, for selected latent widths. The standard autoencoders trained at increasing sizes (Figure 3, A) show increasing reconstruction fi-delity, with recognizable digits appearing around 4-6 latent features. When removing features from the size 32 autoencoder (Figure 3, B), the encoding are clearly corrupted unless most features are present. PCAAE (Figure 3, C and D) has only similar reconstructions to the standard autoencoders once sufficient features are present, and can also content with significant ablations. Finally, Trian-gular Dropout (Figure 3, E) has similar reconstructions to the individual autoencoders, even though its latent size is determined by ablation. A user of this model could select some latent size approxi-mately between 6 and 32 and expect it to yield recognizable digits.

We also explored the application of Triangular Dropout to the CelebA dataset, and find similar effects. This model had a latent size of 128. Full architectural and training details are available in the Appendix. As seen in Figure 3 F, the most compressed features of the autoencoder also capture the most significant features, with additional detail captured by subsequent outputs. We also noted that Triangular Dropout may have unusual behavior with batch normalization (Ioffe & Szegedy, 2015) or variational encodings (Kingma & Welling, 2014), which shift or redefine outputs with a value of zero. These artifacts are primarily seen when width is severely reduced. Reconstructions using these architectures are available in the Appendix.

## 5 PARAMETER REDUCTION IN LARGE NETWORKS

The results of our MNIST autoencoding experiments demonstrate the primary property of a Trian-gular Dropout layer: it can be reduced in width after training and still provide meaningful output. We now consider the ramifications of such a mechanism in the supervised learning domain.

Many powerful models for image feature extraction and natural language are so large that they cannot be easily utilized by the average practitioner at their full sizes. To make them more accessible and study the effects of scale, multiple specific sizes are often trained. For example, VGG (Simonyan

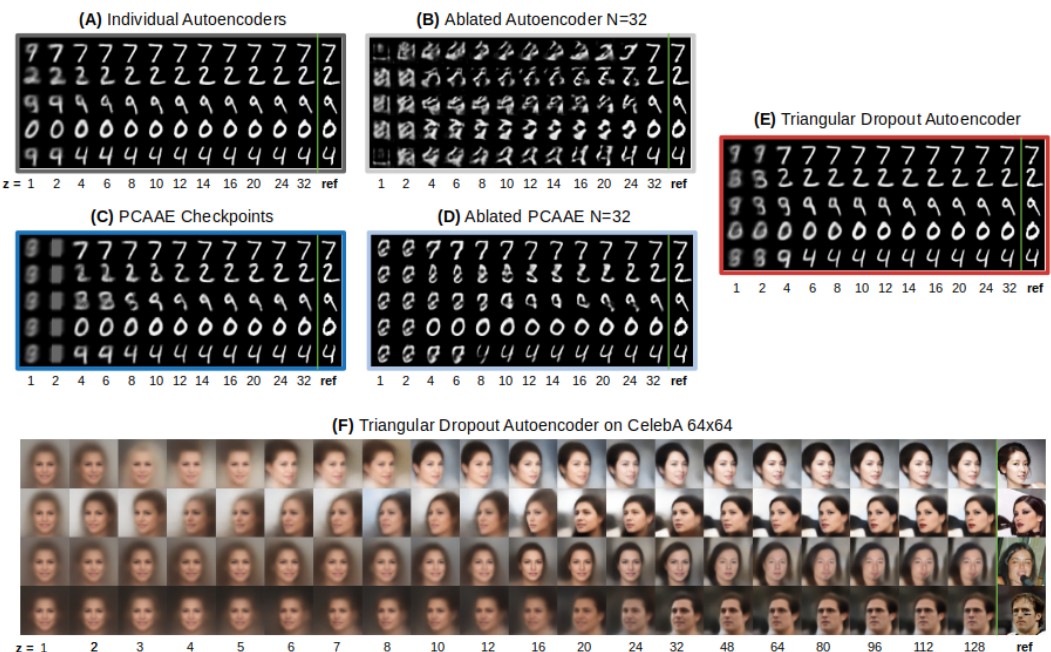

Figure 3: Qualitative analysis of MNIST reconstructions for all methods. Colored borders correspond to line color in Figure 2. **A:** Reconstructions from individual autoencoders at selected sizes, with original inputs on the right. **B:** Width 32 autoencoder with latent features ablated to desired size. **C:** Reconstructions from PCAAE checkpoint models produced by progressively training latent channels towards a full width of 32. **D:** Width 32 PCAAE with latent features ablated to desired size. **E:** Reconstructions from Triangular Dropout trained once at size 32, with smaller size encodings created via ablation.

& Zisserman, 2015) is often used at one of 4 sizes (VGG11, VGG13, VGG16, and VGG19), and so is GPT-2 (Radford et al., 2019) (sizes small, medium, large, and extra large). We envision Triangular Dropout as a stepping stone towards single master models that can be reduced in size arbitrarily to fit the needs of any given user. In this manner, Triangular Dropout could be considered a pruning technique in which the model is trained in preparation for structured pruning.

To motivate this view, we retrain the fully connected classification portion of VGG19 on ImageNet using Triangular Dropout for the two hidden layers of size 4096. We then investigate the fidelity of the model when run at reduced widths. We concentrate on this portion of the model because it is fully-connected and contains the majority of VGG's parameters (roughly 123 million out of 143 million in total). Specific training details are available in the appendix.

## 5.1 Model Accuracy Findings

VGG19 retrained with Triangular Dropout does not achieve quite the accuracy of the original VGG19, achieving 69.7 percent accuracy when not reduced in width, compared to 72.4 percent accuracy of the original VGG19 model. This may or may not be significant depending on the application; this would be considered a significant difference in an often-used benchmark like ImageNet, but in a unique real world problem may be acceptable in exchange for the properties that Triangular Dropout imparts to the network.

Additionally, the success of the model with two layers of Triangular Dropout indicates that Triangular Dropout layers can potentially be stacked in sequence to create a multi-layer MLP with adjustable width. In Section 6 we investigate a network with 3 hidden layers, all of which utilize Triangular Dropout.

## 5.2 REDUCED WIDTH

Our primary finding is that Triangular Dropout allows the classification portion of VGG19 to be reduced in width after training, resulting in a dense sub-network that trades accuracy for narrowness. This a known tradeoff from prior work in network pruning (see pruning results for VGG16 on ImageNet in Blalock et al. (2020), figure 3), which Triangular Dropout makes available without introducing sparsity, requiring fine-tuning, or iterative training. Our results using full-batch Triangular Dropout (batch size 4096) are summarized in Table 1 and graphically in Figure 4 with the curve in bold. As seen in this plot, an accuracy of around 50 percent can be achieved with only 1 percent of the original classifier parameters. This corresponds to hidden layers that are roughly 10 percent as wide as in the full model. The hidden layers can be reduced in size by half and only incur a 0.1 percent loss in classification accuracy. This corresponds to ablating more than half of the classifier parameters, and roughly 46 percent of all the parameters in VGG19.

## 5.3 EFFECTS OF BATCH SIZE

Since training with such a large batch size is not always practical for large image models, we also explore training with batch sizes that are fractions of the full-size batches. In this setting, the triangular mask takes the form of a block matrix with blocks of size 1 row by d columns, where d = 4096/B. The width ablation curves are shown in Figure 4. The primary result is that the full-size batches appear the most stable and perhaps best-performing, but all batch sizes have similar trends of accuracy vs width. Notably, this tradeoff does seem to begin breaking down with very small batches, as seen by batch sizes 128 and 256 having irregular curve profiles.

| Ablated Width | Accuracy | Classif. Params | Param. Reduction |
|---|---|---|---|
| VGG19 Reference | 72.4% | 123,642,856 | N/A |
| 4096 (full) | 69.7% | 123,642,856 | 0.0% |
| 2048 | 69.6% | 57,627,624 | 53.4% |
| 1024 | 69.2% | 27,765,736 | 77.5% |
| 512 | 67.8% | 13,621,224 | 89.0% |
| 256 | 65.3% | 6,745,576 | 94.5% |
| 128 | 61.7% | 3,356,904 | 97.3% |
| 64 | 54.9% | 1,674,856 | 98.6% |
| 32 | 38.8% | 836,904 | 99.3% |

Table 1: Accuracy and parameter numbers for VGG19 trained with Triangular Dropout, with various layer widths after training. The first row shows reference values for the original VGG19 architecture.

## 6 COMPRESSED RL POLICIES

In this section, we explore the application of Triangular Dropout to sequential decision making. Specifically, we first train RL policies on four locomotion control tasks from the MuJoCo (Todorov et al., 2012) environment suite within OpenAI Gym (Brockman et al., 2016). We then then distill the learned policies into secondary networks that use Triangular Dropout, either in a single hidden layer or for all hidden layers. We examine the performance of these distilled policies for different levels of width reduction, providing measures for both the compressibility of these policies and the complexity of the tasks they solve. Finally, we select a compressed architecture that still performs well for each task and evaluate its performance when retrained from scratch.

We examine four environments: Hopper-v3, HalfCheetah-v3, Walker2d-v3, and Ant-v3, all of which come from OpenAI Gym's MuJoCo environment collection. We selected these tasks because they have similar objectives (locomote forward at the highest speed possible), but may differ in complexity.

## 6.1 POLICY DISTILLATION WITH TRIANGULAR DROPOUT

We first determine if Triangular Dropout can be used to create variable-width policy networks, and compare the performance of the ablated policies to the original experts. We also wish to explore if the compressibility of the policy (how much it can be reduced in width without severely affecting

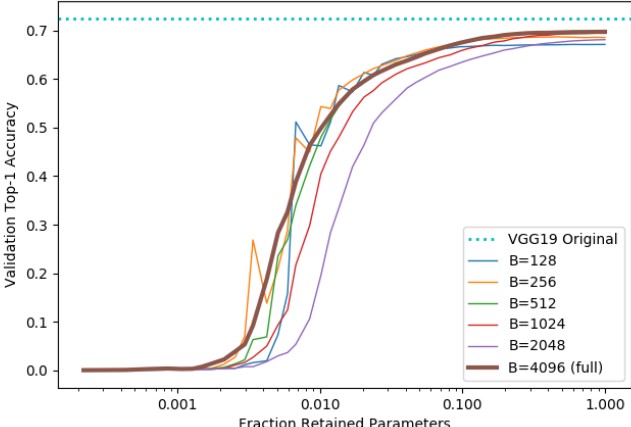

Figure 4: Validation accuracy of VGG19 with Triangular Dropout as the two hidden layers in the classification MLP are modified in width after training. Several batch sizes are tested, reduced from the full-sized square batch (bold). For comparison, the reported validation accuracy of the original VGG19 is also shown (teal).

performance) is a potential metric of task complexity, an important and open problem in reinforcement learning, multitask learning, and meta-learning. We do not use Triangular Dropout in RL directly because the masking changes the policy's behavior during data collection.

For each environment, we first train an expert policy (three fully-connected layers of size 48) using proximal policy optimization (PPO) (Schulman et al., 2017), and then imitate them through supervised learning. To imitate these expert policies, we first gather (state, action) pairs from several thousand evaluation runs with occasional random actions taken to prevent over-sampling of the most likely trajectories. We randomly down-select to 100,000 data points and train a new model to imitate this policy in a supervised fashion. The new model consists of 3 fully-connected layers of size 48, with either the middle layer using Triangular Dropout, or all layers. After training, we can reduce the width of the Triangular Dropout layers to enforce a compression in the mapping of states to actions. Training details of both the experts and imitation networks are available in the Appendix.

We examine the performance of the ablated policies by finding the smallest layer widths that maintain at least 90% performance (speed) compared to the original experts. We measure performance in terms of average top speed over 100 episodes, which is a smoother metric than raw reward (and is the primary goal of the environments). We find that in the case with a single middle Triangular Dropout layer, the policies for Hopper, HalfCheetah, Walker2d, and Ant can be compressed to middle bottlenecks of 8, 2, 2, and 7 respectively. For imitation networks consisting only of hidden Triangular Dropout layers, we find widths of 8, 6, 6, and 9 recover at least 90% of the expert performance, respectively. This implies that Hopper and Ant may be more complex than HalfCheetah and Walker, since they seem to require more parameters in their solutions. It is also interesting that HalfCheetah and Walker2d can be performant with a bottleneck of size 2, which is far less than the action space of these environments (size 6). As described in Paulhamus et al. (2019), this may imply that the solutions reside in a two-degree-of-freedom subspace of possible actions.

## 6.2 COMPARISON TO RETRAINED ARCHITECTURES

Finally, we take the 90% performance ablated architectures and retrain them from scratch in the same manner as the original experts. Here the reduced Triangular Dropout layers have been replaced with fully-connected layers of the reduced size. Comparing the retrained experts with the original expert policies, the retrained experts had comparable (and in some cases better) top speeds to the originals, despite their vastly reduced network size. Compared to the reduced networks with Triangular Dropout (before retraining), there are cases in which retraining yields performance gains,

but this was not always true. For some problems simply reducing the policy width after imitating with Triangular Dropout will maintain performance.

A detailed summary of the ablation experiments in this section are shown in Figure 5. RL training curves and hyperparameters are available in the appendix, along with numerical values for top speed and average reward of all policies.

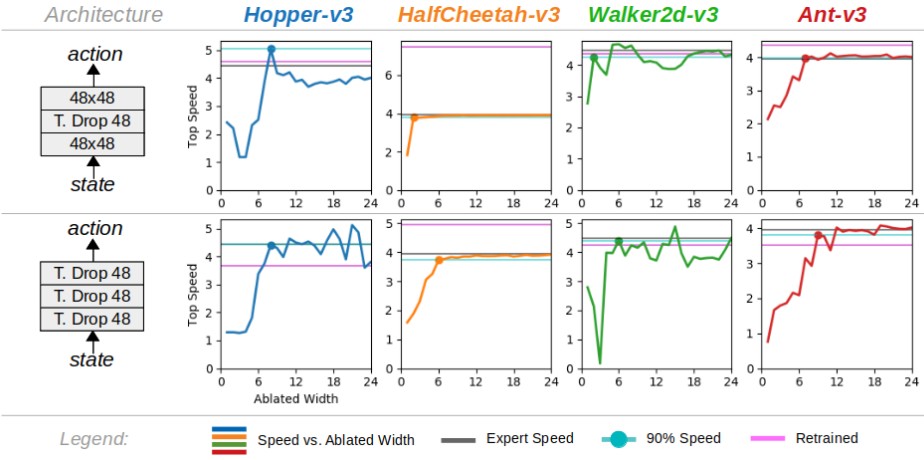

Figure 5: Curves describe imitator performance on each environment as the Triangular Dropout layers are reduced in width after training, using architectures shown on the far left. We only show widths 1 through 24 for readability. Horizontal lines indicate reference performance of original experts (black), ablated policies with at least 90% expert performance (teal), and retrained policies using the compressed architectures (magenta).

## 7 DISCUSSION

Throughout our experimentation, we found several characteristics of Triangular Dropout that were surprising. Firstly, the results indicate that features learned with Triangular Dropout are indeed one-way dependent. That is, the $n + 1th$ feature relies on the $nth$ to be meaningful, but not the other way around. It is not immediately clear that this property should form inherently, since for a given training batch there are many examples for which adjacent features are both optimized.

Perhaps our most significant finding was that Triangular Dropout seems to encourage compression in learned features, enabling structured pruning after training. In many cases the network used far less of its available width than we had allocated, and to our surprise did not spread representation over all available resources. We believe Triangular Dropout to be an interesting mechanism that could lead to networks which are self-organizing and variable in size. In addition to the useful properties discussed above, a network that can be pruned to dense sub-networks without retraining has many practical benefits, especially as AI models get larger and mobile and embedded platforms become increasingly widespread.

## 8 FUTURE DIRECTIONS

There are several promising directions in which to continue this work. Triangular Dropout has only been designed and tested for fully-connected layers, and immediate future work could examine reforming Triangular Dropout for convolutions or transformers. Most prior work in structured pruning has focused on convolutions. Additionally, Triangular Dropout allows a network layer to be resized in width, but not depth. A corresponding mechanism which would allow for variable-depth networks post-training would be similarly useful, and very powerful when combined with variable width. Investigating solutions for applying Triangular Dropout directly to on-policy RL would also be worthwhile, as we believe our technique holds promise for developing compressed policies.

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

# A APPENDIX

## A.1 AUTOENCODER EXPERIMENTS

In this section we detail architectures, training routines, and additional experiments concerning Triangular Dropout applied to MNIST autoencoders.

### A.1.1 MNIST ARCHITECTURE AND TRAINING DETAILS

For all MNIST cases the network architectures were as follows, given some encoder output size $z$:

| Model | Type | Activation | Parameters |
|---|---|---|---|
| **Encoder** | Convolution | Relu | 32 Filters, 9x9 Kernel, Stride 1 |
| | Convolution | Relu | 32 Filters, 7x7 Kernel, Stride 1 |
| | Convolution | Relu | 32 Filters, 3x3 Kernel, Stride 1 |
| | Fully-Connected | linear | 4608 Inputs, $z$ Outputs |
| **Decoder** | Fully-Connected | Relu | $z$ Inputs, 4608 Outputs |
| | Transp. Conv. | Relu | 32 Filters, 3x3 Kernel, Stride 1 |
| | Transp. Conv. | Relu | 32 Filters, 7x7 Kernel, Stride 1 |
| | Transp. Conv. | Relu | 32 Filters, 9x9 Kernel, Stride 1 |
| | Convolution | sigmoid | 1 Filter, 3x3 Kernel, Stride 1, Padding 1 |

Models are trained with batch size 1024 using the Adam optimizer, with a learning rate of 0.005 that decreases by a factor of 10 if the average epoch loss does not decrease by at least 2 percent over the last 15 epochs. This is repeated for 5 decreases of the learning rate. MNIST images were normalized to values between 0 and 1, and binary cross entropy was used as the loss for reconstruction.

### A.1.2 AVAILABLE WIDTH VERSUS COMPRESSION

We additionally performed experiments to test if the available encoding size affected the learned representations in Triangular Dropout. If extra width is available, does the model utilize that width or is it ignored? In Figure 7 we show a plot of autoencoders trained using Triangular Dropout with encoding widths up to 32, however they were trained with available widths of 4, 8, 16, 32, 64, 128, 256, and 512.

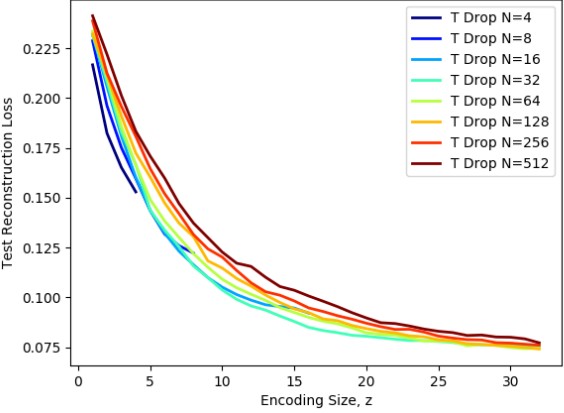

Figure 6: MNIST reconstruction loss for the widths 1 through 32 of autoencoders trained with Triangular Dropout and a variety of available latent widths.

We find that there is a clear pattern of models with more available width being less compressed, but the overall difference in performance diminishes as available width increases.

### A.1.3 CelebA Architecture and Training Details

We use the following architecture for our CelebA experiments. The VAE results described below simply use two parallel encoding layers for the mean and variation, respectively. Models were trained for 100 epochs with batch size 128. We used the Adam optimizer with learning rate 0.001, decreasing by a factor of 10 every 30 epochs.

| Model | Type | Activation | Parameters |
|---|---|---|---|
| **Encoder** | Convolution | Relu | 128 Filters, 5x5 Kernel, Stride 2, Optional BN |
| | Convolution | Relu | 128 Filters, 5x5 Kernel, Stride 2, Optional BN |
| | Convolution | Relu | 64 Filters, 3x3 Kernel, Stride 1 |
| | Fully-Connected | Relu | 7744 Inputs, 2048 Outputs |
| | Fully-Connected | linear | 2048 Inputs, $z$ Outputs |
| **Decoder** | Fully-Connected | Relu | $z$ Inputs, 2048 Outputs |
| | Fully-Connected | Relu | 2048 Inputs, 7744 Outputs |
| | Transp. Conv. | Relu | 128 Filters, 3x3 Kernel, Stride 1, Optional BN |
| | Transp. Conv. | Relu | 128 Filters, 5x5 Kernel, Stride 2, Output Padding 1, Optional BN |
| | Transp. Conv. | Relu | 128 Filters, 5x5 Kernel, Stride 2, Output Padding 1 |
| | Convolution | Relu | 128 Filters, 3x3 Kernel, Stride 1, Padding 1, Optional BN |
| | Convolution | Relu | 128 Filters, 3x3 Kernel, Stride 1, Padding 1, Optional BN |
| | Convolution | Relu | 128 Filters, 3x3 Kernel, Stride 1, Padding 1 |
| | Convolution | sigmoid | 3 Filter, 3x3 Kernel, Stride 1, Padding 1 |

### A.1.4 CelebA Reconstructions with Alternative Architectures

Please see reconstruction examples on the following page.

**(A)** Triangular Dropout Autoencoder

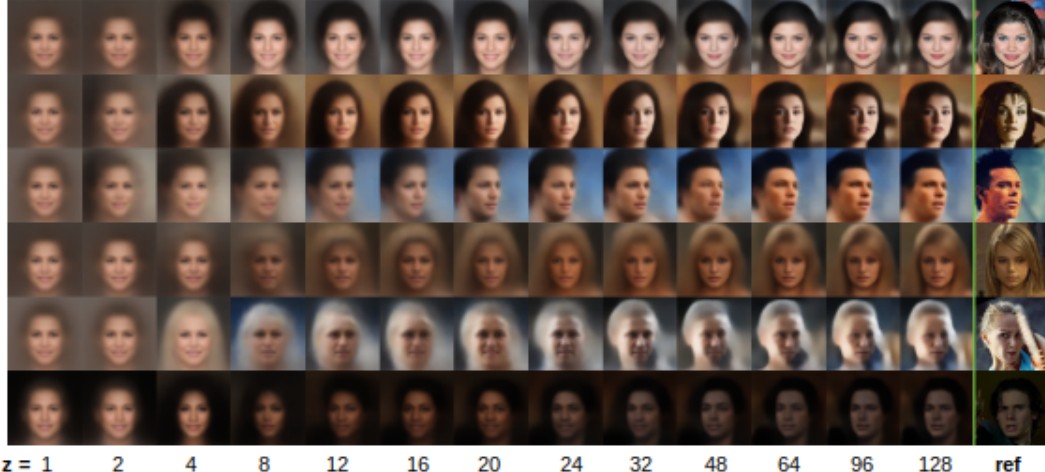

**(B)** Triangular Dropout Autoencoder with Batch Normalization

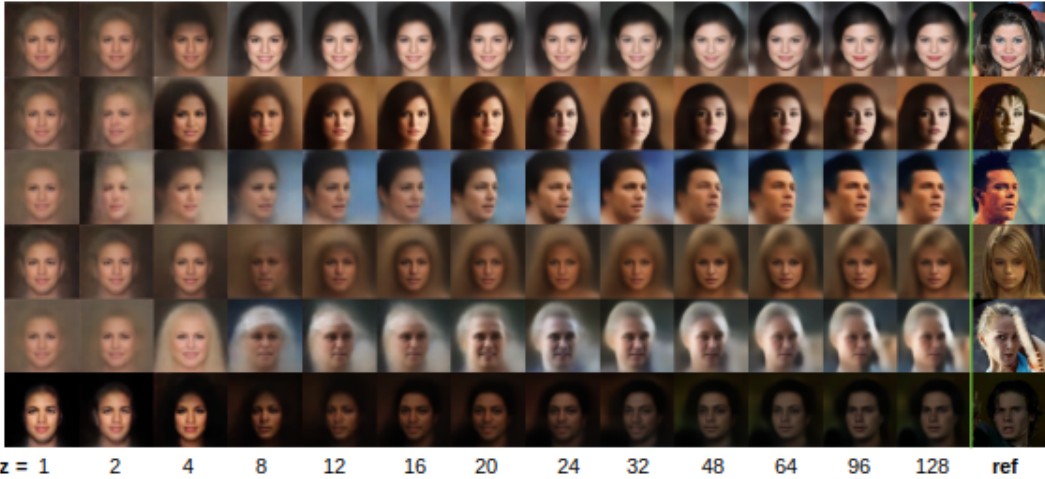

**(C)** Triangular Dropout Variational Autoencoder (VAE)

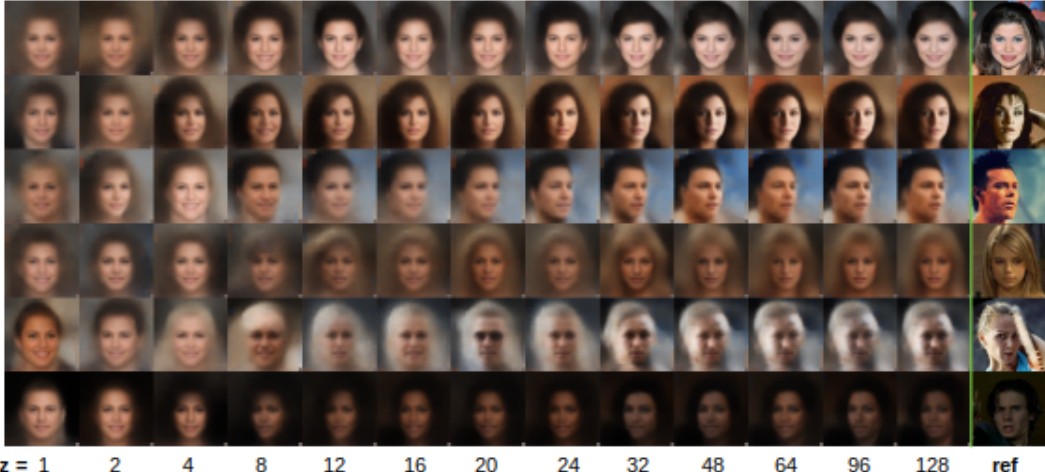

Figure 7: Reconstructions of previously unseen CelebA images on various architectures using Triangular Dropout at the encoding layer.

## A.2 VGG19 Training Details

Our version of VGG19 with Triangular Dropout was created by retraining the classification portion of VGG19. The convolutional layers were frozen and not retrained. The convolutional layers output 25088 features for a given input, which then pass through two hidden layers of size 4096, and finally output as 1000 features to be interpreted as classification scores for the image. Our model simply replaced the two hidden layers with Triangular Dropout layers of the same size.

We trained our model for 100 epochs using a batch size of 4096 in order to apply a full triangular mask to the batch. We used SGD with an initial learning rate of 0.01, reducing the learning rate by a factor of 10 every 30 epochs.

It is unclear if our model would have behaved differently if the entire architecture were retrained, which may have enabled Triangular Dropout to be applied to the feature descriptors of length 25088. This is of interested because VGG19's 25088 features are often used as a starting point for other image processing tasks.

## A.3 RL Training Schemes

In this section we detail hyperparameters and training routines needed to reproduce our experiments from Section 6.

### A.3.1 PPO Hyperparameters and Training Details

The PPO expert discussed in Section 6 consisted of separate policy and value networks. The networks were MLPs with three hidden layers of size 48 (or less for the retrained policies), with input and output sizes determined by the environment. Value networks consisted of two layers of width 128. All layers (besides value output) used hyperbolic tangent as the activation function. We used the implementation of PPO from Stable Baselines 3 (Raffin et al., 2019).

| Hyperparameter | Value | Hyperparameter | Value |
|---|---|---|---|
| Training Steps | 2 million | Entropy Coefficient | 0.0 |
| Parallel Actors | 1 | Clip Range | 0.2 |
| Rollout Length | 2048 | Adam Learning Rate | 0.0003 |
| Training Epochs | 10 | Gamma, $\gamma$ | 0.99 |
| Value Coefficient | 0.5 | Lambda, $\lambda$ | 0.95 |

Table 2: PPO Hyperparameters

### A.3.2 PPO Learning Curves and Statistics

For reference, we provide the learning curves and final performance of our PPO agents. Please reference the figures on the subsequent pages.

### A.3.3 Imitation Network Training Details

The imitation networks consisted of MLPs with 3 hidden layers of width 128, the middle layer using Triangular Dropout. Internal layers used the Relu activation function while the final layer used hyperbolic tangent. The networks were trained in a supervised fashion on 100,000 (state, action) pairs collected from running the PPO-trained policies on their respective environments. The networks were trained for 240 epochs with a batch size of 1024, using the Adam optimizer with a learning rate of 0.01. Every 80 epochs, the learning rate was decreased by a factor of 10.

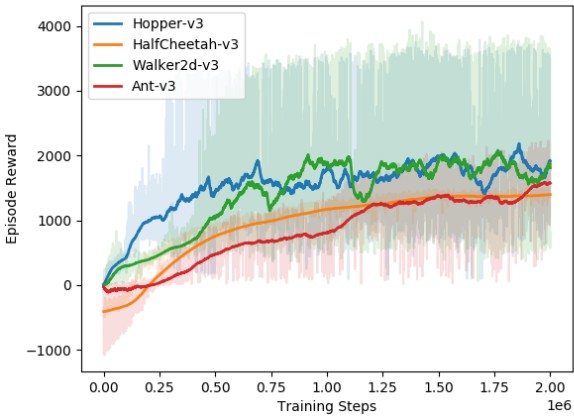

Figure 8: PPO training curves for expert policies

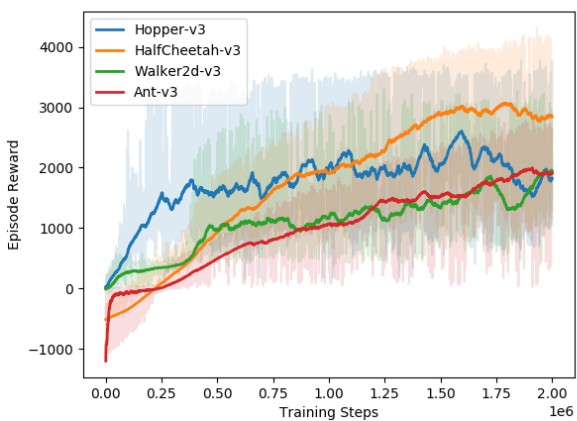

Figure 9: PPO training curves for retrained policies having a compressed middle layer.

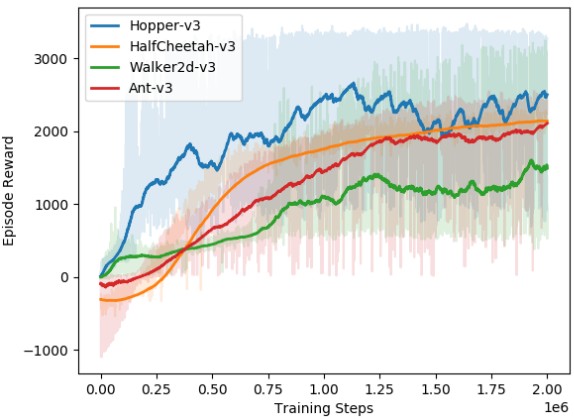

Figure 10: PPO training curves for retrained policies having all layers compressed.

| | Environment: | *Hopper-v3* | *HalfCheetah-v3* | *Walker2d-v3* | *Ant-v3* |
|---|---|---|---|---|---|
| **Original Experts** | Top Speed | 4.44 ± 0.76 | 3.95 ± 0.05 | **4.47 ± 0.50** | 3.95 ± 0.42 |
| | Reward | **2812 ± 944** | 1411 ± 47 | 1861 ± 1189 | 1570 ± 576 |
| | Architecture | 48 / 48 / 48 | 48 / 48 / 48 | 48 / 48 / 48 | 48 / 48 / 48 |
| **90% Speed Compressed (Middle)** | Top Speed | **5.05 ± 0.25** | 3.79 ± 0.04 | 4.26 ± 0.40 | 3.97 ± 0.30 |
| | Reward | 1353 ± 297 | 1442 ± 57 | 737 ± 255 | 1867 ± 590 |
| | Architecture | 48 / 8 / 48 | 48 / 2 / 48 | 48 / 2 / 48 | 48 / 7 / 48 |
| **90% Speed Compressed (All)** | Top Speed | 4.42 ± 0.32 | 3.73 ± 0.03 | 4.40 ± 0.48 | 3.80 ± 0.27 |
| | Reward | 1051 ± 261 | 1365 ± 89 | **1990 ± 832** | 2028 ± 467 |
| | Architecture | 8 / 8 / 8 | 6 / 6 / 6 | 6 / 6 / 6 | 9 / 9 / 9 |
| **Retrained (Middle)** | Top Speed | 4.60 ± 0.66 | **7.49 ± 0.23** | 4.37 ± 0.71 | **4.37 ± 0.56** |
| | Reward | 2356 ± 984 | **3318 ± 947** | 1826 ± 720 | 2099 ± 787 |
| | Architecture | 48 / 8 / 48 | 48 / 2 / 48 | 48 / 2 / 48 | 48 / 7 / 48 |
| **Retrained (All)** | Top Speed | 3.68 ± 0.61 | 4.95 ± 0.25 | 4.25 ± 0.49 | 3.53 ± 0.18 |
| | Reward | 2422 ± 1031 | 2150 ± 90 | 1460 ± 878 | **2208 ± 479** |
| | Architecture | 8 / 8 / 8 | 6 / 6 / 6 | 6 / 6 / 6 | 9 / 9 / 9 |

Figure 11: Average top speed, average reward, and architecture details for all agents in the RL experiments.

