# OpenReview forum: "Triangular Dropout: Variable Network Width without Retraining"
_ICLR.cc/2022/Conference — ICLR 2022 Submitted_

### Official Review · Reviewer_yaJC · 2021-10-20

**Correctness:** 3
**Technical Novelty And Significance:** 3
**Empirical Novelty And Significance:** 1
**Recommendation:** 1
**Confidence:** 5

**Main Review:**

This paper presents a clever idea that is very relevant to practical applications of neural networks, and could potentially impact industrial applications that require tight or variable control over the speed / performance tradeoffs of the system.
Unfortunately, as it stands it fails to provide the level of experimental validation that would be required to make the case for the usefulness of the approach. It's unfortunate because the fundamentals of the method are strong and well-motivated.
There are a number of schemes one could imagine using to generate a progressively larger model: PCAAE is a good baseline, Progressive Networks could be another (https://arxiv.org/abs/1606.04671), but in the end the ceiling is to retrain a different model each time. It's important to characterize the cost of using this scheme against that ceiling, and the paper attempts to do so. However the choice of benchmarks is problematic:
- MNIST is an ok place to start to ascertain that the method isn't broken, no issues here.
- VGG is not a good benchmark to evaluate complexity vs performance. It is notoriously very far from any complexity-vs-performance optimality curve. You can do pretty much *anything* to a VGG network, and get a 10x improvement in complexity. The vast majority of network connections in a VGG net are useless, and can be pruned in any number of ways. So this benchmark is completely uninformative in ascertaining whether this scheme incurs a significant penalty compared to retraining in any practical application. I would urge the authors to pick any family of models that are known to constitute a strong complexity vs. accuracy tradeoff (MobileNets, EfficientNets, ...), and look at how this dropout scheme fares against full retraining. It doesn't have to beat them to be worthwhile, but anyone wanting to use this scheme will want to know what the gap looks like.
- MuJoCo baselines are also notoriously insensitive to network capacity above a threshold, and collapse rapidly below it. Any practical user of those baselines would hence either never use a model that's too small to solve the task, and be completely insensitive to parametric complexity above that threshold. Since the proposed scheme doesn't speed up training over using a large network, there is no practical interest in this scheme here. Because of the overall insensitivity (shown by the step function in Fg 5), we can't extract from this experiment how well this scheme works compared to naive retraining.
Learning whether the effective gap in performance between this scheme and full retraining of a SOTA model is small would be a big deal, and characterizing that gap well would make the difference between a clever, not-well-qualified idea and a great paper.

**Summary Of The Paper:**

This paper proposes a modification to the popular dropout scheme which enables networks to be resized without having to retrain them. The key idea is to dropout units conditionally on others in a sequential fashion, so that a natural importance ordering is induced over each uint of a MLP layer, and no co-dependencies are created between less important units and more important ones, allowing the less important ones to be pruned.

**Summary Of The Review:**

Sound idea, with good potential, but experimental validation lacking.
EDIT POST REBUTTAL: Downgraded to strong reject, high confidence. Merely acknowledging the gaps in validation in the paper doesn't address concerns about experimental validation.

---

### Official Review · Reviewer_8MmW · 2021-10-25

**Correctness:** 4
**Technical Novelty And Significance:** 3
**Empirical Novelty And Significance:** 3
**Recommendation:** 6
**Confidence:** 5

**Main Review:**

**Positives**

- The paper is very well written. Ideas are conveyed clearly, with useful diagrams and math. Experimental choices are properly explained and justified.
- The method is simple and should be easy to adopt by practitioners.
- Experiments are well designed and prove the authors' hypotheses.


**Concerns**

- Although I flagged the experimental design as one of the strengths of the paper because I believe they clearly show the authors' points, it is also true that most experiments are quite toyish (MNIST) or not entirely relevant given the current trends in deep learning (most modern architectures, such as ResNets, do not have fully-connected layers). The paper would be much stronger if it showed the benefits of Triangular Dropout on more realistic settings, and I believe that GAN or VAE experiments similar to the ones in the PCAAE paper would be a good way to show this.
- The background / related work draws quite interesting connections with existing approaches, but in my opinion it should also cover the relationship to the field of neural networks with adaptive computation (which has been mostly developed in the context of RNNs). Some of these works are related to directions mentioned in the future work section such as variable depth (ACT, S-ACT, PonderNet, SkipRNN). There are other works that consider variable width for RNNs, such as Jernite et al. (2017). Please find relevant references below.
- Authors seem to assume that $B \geq n$ so that they can create a lower triangle dropout matrix. This can be problematic for many models used in practice, such as the VGG-19 architecture in the ImageNet experiments, as one may not be able to allocate thousands of examples per batch due to hardware or optimization constraints. However, if I understood the method correctly, the only real requirement is for every row in the dropout matrix to be composed of $p$ 1's followed by $n-p$ 0's (so that one can just sample $p \sim Uniform[0, n-1]$ for every row). I would appreciate if authors could clarify this, and would suggest including this in the paper should it be correct.

**Missing references**

ACT
>Graves. "Adaptive computation time for recurrent neural networks". arXiv preprint arXiv:1603.08983, 2016.

S-ACT
>Figurnov et al. "Spatially adaptive computation time for residual networks." CVPR 2017.

PonderNet
>Banino et al., Pondernet: Learning to ponder. arXiv preprint arXiv:2107.05407.

SkipRNN
>Campos et al., "Skip RNN: Learning to skip state updates in recurrent neural networks." ICLR 2018.

VCRNN
>Jernite et al., "Variable computation in recurrent neural networks". In ICLR, 2017.

**Minor comments / questions**

- How are the autoencoder ablations in Figure 2 performed? Do you just use the first $n$ units in the bottleneck? Have you tried optimizing/searching for the binary mask with $n$ elements that provides the best performance?
- Have you visualized the training curves of the autoencoder with Triangular Dropout? Does it need more iterations to converge than a standard autoencoder?
- The VGG experiments use a huge batch size, which could result in overfitting. Have you trained the same network without Triangular Dropout to rule this out? You may have done this already, I'm unclear about where the accuracy for the baseline VGG-19 comes from (did you retrain it yourselves?).

**Typos**

- Page 3: wrong quotation marks for *"tasks"* (after the equation in the center of the page)
- Page 4, first line: masks -> makes
- Page 5: *ois very similar* -> *is very similar*
- Section 6.1: wrong latex command when citing Kirkpatrick et al.
- Section 6.1: wrong quotation marks before *smallest* (twice in the same line)
- Section 6.2: *Any-v3* -> *Ant-v3*

**Summary Of The Paper:**

This paper proposes Triangular Dropout, a mechanism for training fully-connected layers whose width can be decreased at test time, by using dropout masks with a particular structure at train time. The proposed approach compares favorably to previous methods when training autoencoders of varying bottleneck size on MNIST. Authors show that Triangular Dropout can also be used to tune the parameter count - accuracy trade-off at test time on VGG-19. Finally, the proposed method is used to measure task complexity in RL by using the minimum layer width required to solve the task as a proxy.

**Summary Of The Review:**

This a good paper that proposes a simple yet effective idea and conveys it very clearly. However, I believe that it is currently below the threshold for ICLR due to the scale and relevance of its experiments. I will update my score if authors can provide experiments on more realistic settings (please see my review above for suggestions). I also have some minor concerns regarding the related work section and some technical details that can be solved throughout the discussion period, please see my list of concerns above for more details.


===== POST-REBUTTAL EDIT ====
Updated score from 5 to 6

---

### Official Review · Reviewer_Te1U · 2021-11-02

**Correctness:** 3
**Technical Novelty And Significance:** 2
**Empirical Novelty And Significance:** 2
**Recommendation:** 5
**Confidence:** 4

**Details Of Ethics Concerns:**

In my opinion, the use of facial data (CelebA) should be strongly justified and at the very least the potential concerns and harms associated with it should be discussed. There are many alternatives available to face data.

**Main Review:**

Update after discussion: I increased my overall evaluation after the discussion and related literature has been updated and the conclusions adjusted. I also added a mention to the ethical considerations associated with the use of face data in my review, after the introduction of new results on CelebA data set.

This paper is written in a comprehensive way it is easy to follow, and the description of the experiments and presentation of the results is clear. The main appeal of the proposal, in my opinion, is the simplicity of the technique and I do see potential uses, some of which are described in the paper, such as flexible selection of the amount of resources during inference time. However, the weaknesses I see and that I describe below outweigh these strengths.

In my opinion, one of the main weaknesses of the paper is the lack of an in-depth discussion of the position of this paper within the existing literature. There exists abundant literature on model compression, which is the main use of the proposal, given that this is a topic of great interest for the machine learning industry which currently dominates the field. However, the paper falls short at explaining what gap it fills or what bottleneck it addresses that is missing in the existing literature. For example, there has been a vast amount of work on network pruning, but all that is mentioned in the paper in relation with this section of the literature is in the last paragraph of Section 2 Background, with a couple of sentences arguing how Triangular Dropout is different from pruning. I believe that beyond justifying the distinctive aspects of this paper, the paper would be stronger with a more comprehensive comparison with existing works, explaining which aspects are similar, which are different, which are the common and unique use cases, etc.

Related to this, several of the conclusions from the paper are stated as though they were completely novel and not previously presented in the, for instance, pruning literature. For example, in Section 5.2, the authors discuss that their results show "that there are significant diminishing returns on classifier accuracy with respect to number of parameters". Or later, in Section 6: "The results from Section 5 indicate that VGG19 could use a more compressed architecture and still perform well". This is well-known from the existing literature, but there is no discussion that places these results into a broader context.

The limited novelty could have been compensated by an in-depth investigation of the properties of Triangular Dropout. However, I will argue that the paper also falls short at shedding sufficient light about the proposed idea. I do value that the authors investigated Triangular Dropout in three distinct domains, that is image autoencoders, image classification and reinforcement learning. Nonetheless, the depth of the experimental setup does not seem sufficient to make a strong submission, in my opinion. For example, the autoencoder section (4) is limited to one architecture trained on MNIST, which is currently considered a toy data set by the community; the supervised learning section (5) is limited to the use of Triangular Dropout on the fully connected layers of VGG only. Many relevant questions remain open, which the authors leave for future work: does Triangular Dropout behave similarly on other architectures and data sets? Why is Triangular Dropout not analysed in convolutional layers? Even in Section 5.3 we read that "[i]t is unclear" whether the proposed technique "would have behaved differently if the entire architecture were retrained". In my opinion, this is an important question that a strong paper should give an answer to.

Finally, I believe that some of the conclusions drawn by the authors in the paper go beyond what the experimental setup and results can support. For example, in Section 5.1, the last paragraph states that "the success of the model with two layers of Triangular Dropout indicates that Triangular Dropout layers can be stacked in sequence to create a multi-layer MLP with adjustable width". In order to confidently conclude that Triangular Dropout can be stacked in multiple layers, I believe that the authors should provide more experiments with several architectures (and ideally data sets) where more than two layers are stacked.

I end the review with a series of shorter comments and questions:

* I think a discussion of the _limitations_ of the proposal would make a stronger paper.
* The authors qualify as "small" a drop in performance from 71.3 % to 69.7 % on top-1 accuracy on ImageNet. I believe many in the community would disagree.
* Perhaps a more comprehensive naming of the sections would explicitly mention _autoencoders_, _image classification_ and _reinforcement learning_ in sections 4, 5 and 6, respectively.
* The chosen name for the technique, Triangular Dropout, could be misleading since _dropout_ is widely associated with both _regularisation_ and _random_ dropout of neurons. Triangular Dropout is not aimed at regularisation and is not random, which makes it hard to interpret without further reading.

### Typos

Below are some potential typos or mistakes I have identified during my review.

* Page 4: with a guarantee that the proceeding -> with a guarantee that the preceding
* Page 5: unless the ablation ois -> unless the ablation is
* Page 2: incorrect opening double quotes in "ablation" and "pruning". Check rest of the paper too.

**Summary Of The Paper:**

This paper proposes Triangular Dropout, a technique that multiplies the output of a fully connected layer by lower triangular matrix, with the aim of providing flexible selection of the dimensionality of the layer post-training. The paper analyses the use of this technique on an autoencoder trained to reconstruct MNIST images, on VGG trained on ImageNet and on reinforcement learning tasks.

**Summary Of The Review:**

The paper is well written and presents an interest idea, but the weaknesses outweigh the strengths, to my assessment of the paper. The main concerns are related to the novelty of the proposal and the discussion with existing works, the depth of the experimental setup. I believe that a more in-depth analysis and stronger connections and comparisons with related techniques could make a stronger paper.

---

### Official Review · Reviewer_CzhB · 2021-11-03

**Correctness:** 3
**Technical Novelty And Significance:** 3
**Empirical Novelty And Significance:** 2
**Recommendation:** 5
**Confidence:** 3

**Main Review:**

[Strengths]
- The proposed method is simple yet effective, allowing the models to retain most of their performance when the width is gradually reduced.
- Experiments were extensive, carried out in three different areas, autoencoder, image classification, and reinforcement learning, and well illustrate the potential of the proposed method.

[Weaknesses]
- The experiments successfully illustrate the potential of the proposed method but its usefulness is still in doubt.
1. The autoencoder experiment was carried out on MNIST, which is kind of a toy example; and
2. The image classification experiment was based on VGG19, which is very outdated.

(I do not know much about RL and won't judge the third experiment.)

[Details]
- The mask $M$ seems to be applied elementwise to the initial output $x^TW+b$. In this way, for the equation at page 2 bottom, matrix inner product should be used instead of matrix multiplication. The current formulation might cause confusion.

[Questions]
- Is it true that, after applying the lower-triangular mask, the elements of the output not only depend on different network widths but also utilize different portions of the batch of samples? If yes, is it the case that removing the last column retains most of the performance because parameters in the last column were trained on the fewest samples? I would like to see more discussions on this aspect.

**Summary Of The Paper:**

This paper proposes a method, called Triangular Dropout, to allow a fully connected layer in a network to have variable width at deployment, which is achieved by applying a lower-triangular mask to the output of the fully-connected layer.
Experiments were conducted in three different scenarios, autoencoder, image classification, and reinforcement learning. Results show that models trained with Triangular Dropout retain most of their performance when being ablated.

**Summary Of The Review:**

The research topic is of value and the proposed method is simple and very effective on toy examples.
Leaving Triangular Dropout for convolutions or transformers to future work makes this paper less informative than expected.

---

> ### Comment · Reviewer_CzhB · 2021-12-07
> **Post-rebuttal update**
>
> Thanks very much for adding the experiments on batch size. In fact, I was also expecting some discussions based on the formulation of the methodology itself---but this could be difficult.
>
> I kept my original rating because my concerns about the effectiveness of the proposed method on more realistic scenarios are not well addressed. I am more familiar with CV applications and thus the new version does not relieve my concerns.

---

### Decision · Program_Chairs · 2022-01-20

**Decision:**

Reject

**Comment:**

The submission proposes triangular dropout training to provide adaptive capacity of the network at inference time. The proposed approach is simple and sound. However, the experiments are lacking in terms of complexity of the task and up-to-date architectures (e.g., transformers or convolutional layers) to demonstrate the effectiveness of the method.
Therefore I recommend this paper for rejection.